# Pilot study assessing gut microbial diversity among sexual and gender minority young adults

Ashley Guy[1], Shannon McAuliffe[1], Robbie Cross[2], Yue Zhang[3], Richard E. Kennedy[3], Norman R. Estes[1], Samantha Giordano-Mooga[1], Christine Loyd[1] *

1 Department of Clinical and Diagnostic Sciences, University of Alabama at Birmingham, Birmingham, Alabama, United States of America, 2 Department of Biomedical Engineering, University of Alabama at Birmingham, Birmingham, Alabama, United States of America, 3 Department of Medicine, Division of Gerontology, Geriatrics and Palliative Care, University of Alabama at Birmingham, Birmingham, Alabama, United States of America

* loydcm@uab.edu

**Data Availability Statement:** We have deposited de-identified gut microbiome data in figshare (DOI 10.6084/m9.figshare.25639596). This data will be published and made public upon acceptance of the

## Abstract

Evidence supports that people identifying as a sexual or gender minority (SGMs) experience minority-related stress resulting from discrimination or expectations of prejudice, and that this is associated with increased mental and physical health problems compared to cisgender heterosexuals. However, the biological mechanisms driving minority-related stress impacts remain unknown, including the role of the gut microbiome. Thus, the aim of this study was to determine the relationship between SGM status and gut microbiome health among young adults attending a 4-year university. To this end, a prospective pilot study was completed in the fall and spring semesters of 2021–22. Self-identified SGMs (N = 22) and cisgender-heterosexuals (CIS-HET, N = 43) completed in-person interviews to provide mental health data and demographic information. Nail and saliva samples were collected at the time of interview to quantify chronic and acute cortisol. Stool samples were collected within 48 hours of interview for microbiome analysis. Assessment of the gut microbiota identified a significant reduction in alpha diversity among the SGM group, even when adjusting for mental health outcome. SGM group showed trends for higher abundance of microbes in phylum *Bacteroidetes* and lower abundance of microbes in phyla *Firmicutes*, *Actinobacteria*, and *Proteobacteria* compared to the CIS-HET group. These findings support that the gut microbiome could be contributing to negative health effects among the SGM community.

## Introduction

Individuals identifying as a sexual or gender minority (SGM), including lesbian, gay, bisexual, or transgender, often expect prejudice and experience discrimination as a result of their minority-status at interpersonal, institutional, and societal levels [1,2]. Not surprisingly, the impact of discrimination on the mental and physical well-being of SGMs is profoundly negative. Abundant research supports that SGMs, including young adult SGMs, have an increased

paper for publication. Since the microbiome data is not self-reported data, the authors believe that this data is less likely to be linked back to individual participants of this small pilot study so it can be made publicly available. Alternatively, due to the limited sample size of this small pilot study, it is unethical to publicly share self-reported data due to the increased likelihood of linking the data back to the individuals enrolled in this study. Furthermore, because of the unique social experience of the SGM community that includes multiple levels of discrimination in the US South, it is vital that we keep the identities of our participants confidential. De-identified self-reported data is available only upon request from the office of the Research Director in the UAB Clinical and Diagnostic Sciences (CDS) Department. The current point of contact is Keith McGregor, email: kmmcgreg@uab.edu. The limited dataset will remain with the Research Director's office moving forward and this email address can be contacted for external requests for data access for the foreseeable future. The corresponding author will respond to the journal regarding changes to the email address for data access requests in the long-term. The CDS Research Director's office is developing centralized email address to ensure continued ability to making data access requests.

**Funding:** The funding of this project was taken from departmental funds provided to CLoyd and SGiordano-Mooga (Department of Clinical and Diagnostic Sciences at UAB). CLoyd was also given an FDGP award through the UAB Provost's Office and Department of Clinical and Diagnostic Sciences that was used to support this work. All funds were provided through UAB.

**Competing interests:** The authors have declared that no competing interests exist.

prevalence of mental health symptoms and psychiatric disease, as well as substance abuse and high-risk sexual behavior issues, compared to the majority group (cisgender and heterosexual individuals) [3–6]. While the physical and medical manifestation of SGM status is comparatively understudied, some evidence does support that SGMs are more likely to consume a poor diet and are at a higher risk for metabolic diseases including cardiovascular disease and obesity which elevates their risk for other comorbidities [7–9]. Currently, little is known of the biological factors driving SGM-related health disparities. Thus, this investigation aims to begin to understand underlying biological differences between individuals identifying as SGM and those identifying as cisgender heterosexual by assessing diversity of the gut microbiome.

Evidence supports that the cardiovascular, neuroendocrine, and immune physiological systems can all be activated due to discrimination resulting in a systematic inflammatory state [10]. Importantly, if an individual is exposed to discrimination over long periods of time this can lead to chronic inflammation which can greatly impact health and well-being [11]. Additionally, discrimination is linked to psychological and physiological stress. During the stress response, the sympathetic nervous system (SNS) and the hypothalamic-pituitary-adrenal (HPA) axis are activated resulting in release of neurotransmitters and hormones, including cortisol, which lead to a systemic physiological response that normally promotes survival. When stress is chronic, the response becomes pathological and can promote medical disease and mental health decline [12–15]. There is strong evidence that discrimination promotes the stress response and this contributes to negative health impacts of discrimination [16].

Recent research has focused on understanding how the stress response affects the gut microbiome including diversity of microbial communities, relative abundance of types of bacteria, and function [17,18]. The gut microbiome plays an important role in the body by communicating with the CNS via the gut-brain axis to integrate emotions and cognition with intestinal processes [19]. It is also recognized as a central component to the normal body physiology through its production of short-chain fatty acids and neurotransmitters [20]. Research supports that stress impacts gut microbiota composition resulting in reduced abundance and diversity of different microorganisms along with the reduced integrity of the intestinal barrier leading to increased permeability [21,22]. These changes to the gut contribute to a pro-inflammatory state and gut dysbiosis, both of which have been linked to physical and psychiatric disease [23,24].

Although previous studies have examined perceived stress and the stress response among SGMs, the connection between stress and changes to the gut microbiota has not yet been described to the best of our knowledge. Ultimately, the findings of this pilot investigation will help in beginning to develop a better understanding of modifiable factors contributing to health disparities among sexual and gender minorities.

## Methods

### Study population

Study participants were young adults attending a 4-year university in central Alabama in 2021 and 2022. The recruitment period began on August 18, 2021 and ended on February 1, 2022. The only inclusion criteria to participate was enrollment at the university at the time of the study. Individuals were excluded if they: (1) were under the age of 18; (2) were not enrolled as an undergraduate student at the university; (3) were pregnant; (4) were currently taking antibiotics or had taken them in the 2 weeks prior; and/or (5) had a self-reported diagnosis of a disorder affecting the HPA axis. The study team used word of mouth recruitment as well as flyers to encourage potential participants to complete a screening survey. Screening data was collected using Qualtrics, a web-based survey company (Qualtrics, Provo, UT). The study team

assessed the screening data and contacted eligible potential participants to schedule an interview.

## Ethics statement

Ethical approval was obtained from Institutional Review Board (IRB) of the University of Alabama at Birmingham (IRB-300007479). This study was conducted in accordance to the principles outlined in the Belmont Report. Prior to enrollment in the study, potential participants were informed of study procedures and information about the study in verbal and written form and they provided written consent to participate in the study. All data collected were deidentified and stored in accordance to data privacy and protection standards approved by the University of Alabama at Birmingham IRB.

## Interviews

After enrollment, participants completed a 45-minute in-person interview, which occurred at the start of the semester. Demographic information, psychological data, health data including pharmaceutical medications, and biological samples were collected during the interview. Additionally, participants were asked to collect a stool sample at home and return it to the research team for storage within 48hr following the interview. Trained assessors completed the interviews and collected biological samples.

## Demographic information

Self-reported demographics were collected including age, race, gender identity, and sexual orientation. Gender identity was categorized as male, female, transgender female, transgender male, non-binary, or other. Sexual orientation was categorized as heterosexual (straight), gay, lesbian, bisexual, queer, prefer not to answer, or other. The majority (CIS-HET) group in this study self-reported having a gender identity of male or female and a sexual orientation of heterosexual (straight). All other participants were considered part of the minority (SGM) group (no participants declined to answer). Race included White/Caucasian, Black/African American, Hispanic/Latino, Asian/Asian Indian, Native American/Alaska Native, Middle Eastern/North African, and Native Hawaiian/Other Pacific Islander. Additionally, participants were asked to self-report whether they take neuropsychological medications such as for anxiety or hormone-based medications such as oral contraceptive.

## Psychological measures

**Perceived stress scale-10 (PSS-10).** Perceived stress was quantified using the PSS-10, a 10-question survey that assesses self-reported level of stress experienced over the previous month [25]. Some of the questions asked were: "How often have you been upset because of something that happened unexpectedly? How often have you felt nervous and/or stressed? How often have you felt that you were on top of things?" Reverse scoring was used for 4 of the questions where a response of "very often" indicated lower levels of perceived stress, for the other 6 questions a response of "very often" was indicating higher levels of perceived stress. Scores of 0–13 suggest low perceived stress, 14–26 suggest moderate stress, and 27–40 suggest high stress. The PSS-10 was shown to be a valid measurement of stress among minority groups and college aged young adults [26,27].

**Patient health questionnaire-9 (PHQ-9).** The patient health questionnaire is a 9-question self-administered tool for assessing depression. It has been validated and is an established method for diagnosing Major Depressive Disorder [28]. Participants were asked questions

about experiencing feelings of depression and anhedonia during the previous 2 weeks. For example, we asked "how often over the last two weeks did you have little interest or pleasure in doing things?" An answer of "not at all" indicated low depression, while a "nearly every day" answer indicated high depression. Total scores of 0–4 suggested low depression, 5–9 mild depression, 10–14 moderate depression, 15–19 moderately severe depression, and 20–27 severe depression. PHQ-9 has been used previously among young adults to assess severity of depression [29].

**Generalized anxiety disorder-7 (GAD-7).**   Anxiety severity was determined with the GAD-7. This 7-question assessment is an established screening method for reliably identifying anxiety disorders [30,31]. Participants were asked questions about their experience with anxiety-related feelings during the previous 2 weeks. For example, we asked "how often over the last two weeks did you feel nervous, anxious, or on edge." An answer of "not at all" indicated low anxiety, while a "nearly every day" answer indicated high anxiety. Total scores of 0–4 suggested low anxiety, 5–9 mild anxiety, 10–14 moderate anxiety, and ≥15 severe anxiety.

## Biological measures

**Cortisol analysis.**   Cortisol in the saliva and nails was quantified to assess acute and chronic physiological stress, respectively. Samples were collected at the start of the interview, which occurred between 12-4PM to account for the diurnal cortisol secretion. Participants were asked not to consume food or drink (except for water) 30 minutes prior to sample collection. Saliva was collected via the passive drool method using saliva collection kits, filling a test tube with 1-2ml of saliva. Saliva samples were stored in a -20˚C freezer. Participants were also asked to clip nails from all ten fingers (or toes if fingernails were too short) and place the clippings into a small bag. To extract cortisol from the nails, they were ground into powder. The procedure for assessing salivary and nail cortisol has been described in detail previously [32]. ELISA kits were used to quantify cortisol levels in saliva and nails using the manufacturer's instructions (Salimetrics LLC, State College, PA).

**Gut microbial diversity.**   Participants were provided with a Zymo Research fecal sample collection kit at the end of the interview (Zymo Research, Irvine, CA). The kit included a container to collect feces and an apparatus for placement of the container over a toilet bowl. After a fecal deposit, participants placed a portion of feces into a specimen transport vial. Return of a fecal sample was required within 48 hours of a participant's interview time. Samples were stored in a freezer at -80˚C until processed.

After thawing the sample, microbe identity was discovered using 16s rRNA sequencing via a process described previously [33]. The sequence data analysis used the illumine MiSeq sequencing platform, and the bioinformatics analysis was completed with QWARP workflow [33]. The Shannon diversity index was used to assess microbial species diversity of the microbiome (alpha diversity), which increases with increasing number of species and the consistency of species abundance. The abundance analysis quantified the number of species of bacteria per phylum. Variability of microbial community composition among samples (beta diversity) from participants in each group was analyzed using the Bray Curtis method, the unweighted uniFrac and the weighted uniFrac methods. The uniFrac measurement of beta diversity uses phylogenetic information to compare samples to identify differences among microbial communities for each group [34]. See the public dataset at DOI 10.6084/m9.figshare.25639596.

## Statistical analyses

The study sample characteristics were described with means and standard deviations for continuous variables and frequencies with percentages for categorical variables. Differences

between groups were analyzed using independent samples *t* tests for continuous variables. Differences between categorical variables were evaluated using Pearson's chi-square tests. Linear regression was utilized to evaluate correlation between PSS-10, GAD-7, PHQ-9 scores, alpha diversity, and sexual and gender minority status.

## Results

A total of 65 participants were enrolled in the study. Study sample characteristics are presented in Table 1. The average age of the study sample was 20 years (SD = 1.66; range = 18–25). Of the 65 participants, 22 (33.8%) participants self-identified as SGM (43 identified as CIS-HET). Fourteen participants identified only as sexual minority (1 as gay, 10 as bisexual, 1 as queer, and 2 as other). Eight of the 22 SGM participants self-identified as a gender minority (1 as transgender, 7 as non-binary) and sexual minority (1 as gay, 3 as bisexual, 3 as queer, and 1 as other). Across the whole cohort, 29 participants identified as White/Caucasian (46%), 18 as African American/ Black (28.6%), 9 as Hispanic (14.3%), 6 as Asian/Indian Asian (9.52%), and 1 as Middle Eastern/North African (1.59%). There was no significant difference in age and race between the SGM and CIS-HET groups. The mean summary scores for PHQ-9 were similar among the SGM group (8.36, SD = 5.36) and the CIS-HET group (8.09, SD = 4.82). Similarly, there was no significant difference between groups for the mean scores for the GAD-7 (SGM: 8.0, SD = 5.7; CIS-HET: 9.12, SD = 5.29) or the PSS-10 (SGM: 19.1, SD = 6.02; CIS-HET: 17.8, SD = 5.83). Furthermore, there was no statistical difference in nail and saliva cortisol levels between groups nor a statistical difference in neuropsychological (e.g., anti-anxiety and anti-depressive medications) and hormone (e.g., oral contraceptive) pharmaceutical usage. Among the CIS-HET group, 8 participants (18.6%) reported using neuropsychological medications regularly, while 7 (31.8%) in the SGM group reported using this type of medication regularly (p = 0.376). Further, 8 participants (18.6%) in the CIS-HET group and 4 (18.2%) in the SGM group reported using hormonal medications regularly (p = 1.000). When the SGM group was separated into sexual minority only (N = 14) and gender/sexual minorities (N = 8),

**Table 1. Characteristics of the study sample.** Differences in variables between the cisgender heterosexual (CIS-HET) group and the sexual and gender minority (SGM) group are represented.

| Characteristic | All | CIS-HET | SGM | P Value | N |
|---|---|---|---|---|---|
| | *N = 65* | *N = 43* | *N = 22* | | |
| Age, Mean (SD) | 20.0 (1.66) | 20.2 (1.70) | 19.5 (1.50) | 0.121 | 61 |
| Race, No. (%) | | | | | |
| African American/Black | 18 (28.6) | 12 (28.6) | 6 (28.6) | | |
| Asian/Asian India | 6 (9.52) | 6 (14.3) | 0 (0.00) | | |
| Caucasian | 29 (46.0) | 17 (40.5) | 12 (57.1) | | |
| Hispanic | 9 (14.3) | 6 (14.3) | 3 (14.3) | | |
| Middle Eastern/ North African | 1 (1.59) | 1 (2.38) | 0 (0.00) | | |
| PHQ-9 Score, Mean (SD) | 8.18 (4.97) | 8.09 (4.82) | 8.36 (5.36) | 0.843 | 65 |
| GAD-7 Score, Mean (SD) | 8.74 (5.41) | 9.12 (5.29) | 8.00 (5.70) | 0.449 | 65 |
| PSS-10 Score, Mean (SD) | 18.3 (5.88) | 17.8 (5.83) | 19.1 (6.02) | 0.427 | 65 |
| Nail Cortisol (SD) | 0.09 (0.23) | 0.06 (0.15) | 0.14 (0.33) | 0.284 | 60 |
| Saliva Cortisol (SD) | 0.58 (0.59) | 0.58 (0.55) | 0.57 (0.69) | 0.939 | 65 |
| Shannon Alpha Diversity, Mean (SD) | 5.17 (0.69) | 5.35 (0.57) | 4.81 (0.78) | 0.034* | 41 |

PHQ-9 = Patient Health Questionnaire 9, GAD-7 = Generalized Anxiety Disorder 7, PSS-10 = Perceived Stress Scale 10. Statistical significance
*P < .05.

there was no significant difference between groups in demographics (age, race), psychological factors (PHQ-9, GAD-7, PSS-10), or nail and saliva cortisol levels measured.

Assessment of gut microbial diversity identified a significant reduction in alpha diversity among the SGM group (4.81, SD = 0.78) compared to the CIS-HET group (5.35, SD = 0.57; p = 0.034). When sexual minorities only were analyzed separately from gender and sexual minorities compared to the CIS-HET group, alpha diversity was significantly reduced among sexual minorities only (N = 14; Shannon Alpha Diversity mean = 4.89, SD = 0.76) and gender and sexual minorities (N = 8; Shannon Alpha Diversity mean = 4.69, SD = 0.89) compared to the CIS-HET group.

To identify whether psychological factors moderated the effect of SGM status on gut microbial diversity, linear regression was used. Linear regression modeling identified lower alpha diversity among SGMs when adjusting for GAD-7 (p = 0.014), PSS-10 (p = 0.018), or PHQ-9 (p = 0.017) scores. Alpha diversity scores for the SGM group were estimated to be 0.55 points lower than the CIS-HET group when adjusting for GAD-7 score, 0.53 points lower when adjusting for PSS score, and 0.54 points lower when adjusting for PHQ score. Beta diversity analysis between the CIS-HET and SGM groups identified a significant difference between groups using the unweighted uniFrac method (p = 0.041). However, no difference was observed using the Bray Curtis and the weighted uniFrac (p = 0.081 and 0.645 respectively).

As illustrated in Fig 1, the dominant microbial phyla across groups were *Firmicutes* and *Bacteroidetes* followed by *Actinobacteria*, *Proteobacteria*, and *Epsilonbacteraeota*. Average relative abundance of *Bacteroidetes* and *Firmicutes* was 39.25% and 56.31% in SGM group and 28.96% and 65.18% in CIS-HET group, respectively. That of *Actinobacteria*, *Proteobacteria*, and *Epsilonbacteraeota* was 2.11%, 1.85%, and .01% respectively among the SGM group and 3.04%, 2.11%, and .002% respectively among the CIS-HET group. Report of the taxonomic breakdown of microbial differences between the CIS-HET and SGM groups can be found in a public dataset (DOI 10.6084/m9.figshare.25639596).

## Discussion

This pilot study aimed to assess the microbial diversity of the gut among sexual and gender minorities (SGMs). Herein, we show that gut microbial diversity was reduced among SGM participants compared to CIS-HET participants, and this difference was independent of self-reported depression, stress, and anxiety levels, as well as acute and chronic cortisol levels. Additionally, the relative abundance of microbial phyla commonly found in the human gut showed trends towards differences between the two groups. Overall, the findings presented herein show that SGM status is associated with disruptions in the bacterial diversity of the gut microbiome.

To the best of our knowledge, this is the first report assessing the health of the gut microbiome among SGM individuals. While it is clear based on published research that psychological stress leads to harmful changes in the gut microbiome including changes in alpha diversity [35,36], little published evidence exists showing the impact of minority stress on gut health. A recent systematic review stated that the authors were unable to assess the influence of basic demographic variables that contribute to minority status (e.g., race, gender, and sexual orientation) on stress-related changes to the gut due to limited data availability [36]. Additionally, others have stated that research on the effect of minority-related stress on the gut microbiome could help in understanding the interaction between health inequity and disease prevalence [37]. This study begins to address this gap by examining differences in the gut microbiome and mental health outcomes of SGMs and CIS-HET young adults. Overall, the evidence supports that SGMs have differences in the gut microbiome that could be contributing to health inequity and risk of disease.

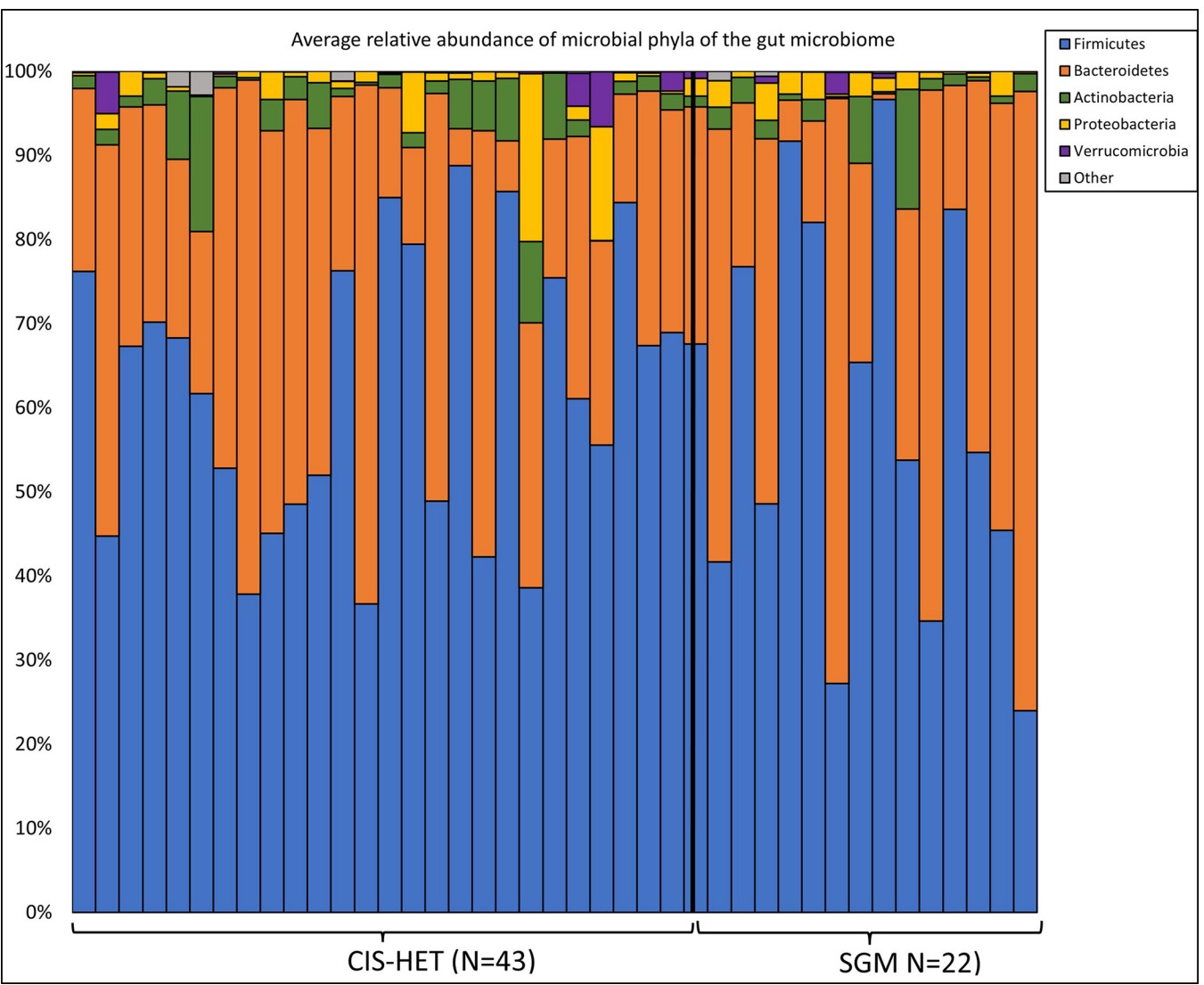

**Fig 1. The average relative abundance of microbial phyla of the gut microbiome among the CIS-HET and SGM groups.** Phyla represented compose >0.001 (>0.1%) of the total microbiome. Phyla that compose <0.001 (<0.01%) are combined and represented as other.

The key finding of this study is our observation that the SGM group had decreased alpha diversity of the gut microbiome compared to the CIS-HET group, and this finding was not driven by psychological variables measured in the study. Evidence supports that reduced gut microbe alpha diversity contributes to gut dysbiosis, or a disruption in the microbiome resulting in an imbalance in the microbial communities present, which is associated with a variety of issues including digestive diseases, psychiatric diseases including those related to mood and cognition, metabolic diseases and immunity-related diseases [38]. This supports that reduced gut microbial diversity could be placing the SGM group at risk of health issues.

Another key finding was that the unweighted uniFrac measure of beta diversity was significantly different between the CIS-HET and the SGM groups. These data suggest some differences in the microbial communities present among the CIS-HET and SGM groups. Assessment of variance in abundance of gut bacterial phyla identified specific phyla trending higher or lower in the SGM group compared to the CIS-HET group, which could be

contributing to negative health impacts. Specifically, 6 species of phylum *Epsilonbacteraeota* tended to be more abundant in SGM group while species in phylum *Bacteroidetes* trended lower in this group. Higher relative abundance of *Epsilonbacteraeota* and lower relative abundance of *Bacteroidetes* is associated with coronary heart disease [39]. Additionally, most of the species of *Firmicutes* that were observed were less abundant in SGM participants, and reduced gut *Firmicutes* is linked to poor mental health outcomes that are common among the SGM community including anxiety and chronic stress [40,41]. Reduced gut *Firmicutes* is also associated with irritable bowel disease and type 1 and 2 diabetes [42]. Larger, more robust, studies are needed to determine the effect of different microbial abundances on the overall health of SGM individuals.

Despite prior studies showing that SGMs experience higher psychological distress and mental health issues [6], we did not observe group-level differences in self-reported anxiety, stress, and depression using the GAD-7, PSS-10, and PHQ-9, respectively. It is possible that some assessments used for this analysis do not accurately measure the outcomes in the SGM population [43]. Alternatively, it is also possible that greater self-acceptance of one's sexuality buffered the negative impact of heterosexism [44]. The university that housed this study has a variety of resources that aim to support the SGM student body, which may have impacted self-acceptance, mental health variables, and cortisol. Another possibility is that SGM participants had biological impacts of mental health issues without self-reported awareness of it due to the observation that they had reduced relative abundance of gut *Firmicutes* without concomitant decline in mental health. Furthermore, we did not observe a difference in acute and chronic cortisol levels between groups, which agrees with the perceived stress findings. It is unclear if other cortisol-independent pathways of stress, such as those driven by epinephrine [45], contributed to the gut microbiome observations in this study. Future research should determine the role of cortisol-independent pathways of stress on gut microbiome health among the SGM community.

This study has a few limitations. First, the population for recruitment was restricted to undergraduates at a 4-year university, which impacted the age range of participants enrolled. Further, the SGM group did not include individuals identifying as lesbian nor did it include gender minority only individuals thus, findings do not apply to these minority groups. Furthermore, it is possible that some SGM individuals were missed because they were not ready to self-report their sexual and/or gender minority identity. Additionally, we did not examine factors that are known to affect the health of the gut microbiome including dietary habits, physical activity, and sleep [46], consequently it is unclear how group differences in these variables impacted study findings. Finally, a convenience sample was selected and the sample size was small for this pilot study, thus our analysis combined all SGM members into a single group, which limited our ability to differentiate impacts on the gut microbiome among distinct sexual and gender minority groups. Therefore, a larger more comprehensive study focused on replicating findings presented herein, identifying differences between SGM sub-groups, and determining the role of non-stress related health conditions on the microbiome is needed.

Despite these limitations, there were noteworthy study strengths to highlight. Specifically, the sample was racially diverse which enhances the generalizability of the findings. Trained study personnel that conducted the interviews were peers of the participants to limit stress caused by the interview and promote accuracy in self-report on mental health variables. Furthermore, stress was measured through self-report and physiological assessment to reduce subjectivity.

## Conclusion

The gut microbiome remains a topic of interest among researchers and medical professionals because of its role in health and disease, and the fact that it is something we can change

through targeted interventions. This report shows that identifying as a sexual and gender minority is associated with a reduction in bacterial diversity of the gut microbiome, which supports that programs for improving gut microbial diversity could positively impact health among SGMs and reduce health inequity in this population.

## Acknowledgments

We would like to thank all participants that volunteered to be in this study. We would also like to thank Casey Morrow at the UAB Microbiome Core for his assistance in microbiome analysis and data curation. Additionally, we want to acknowledge all undergraduate students that were part of the Gut Feelings research team and the UAB School of Health Professions Honors Program. We would like to specifically acknowledge students that played a role in this work but did not meet the threshold of authorship: Laci Turner, Kylie Blue, Alaina Little, Zaurayze Rehman, Lisa Patel, Yug Patel, Lauren Briggs, Gavin Bingham, Sarah Fu, Daniel Phillips, Hannah Alsup, Steve Otero, and Jordan Carson.

## Author Contributions

**Conceptualization:** Norman R. Estes, Samantha Giordano-Mooga, Christine Loyd.

**Data curation:** Ashley Guy, Shannon McAuliffe, Robbie Cross.

**Formal analysis:** Yue Zhang, Richard E. Kennedy.

**Funding acquisition:** Samantha Giordano-Mooga, Christine Loyd.

**Investigation:** Ashley Guy, Shannon McAuliffe, Norman R. Estes, Samantha Giordano-Mooga, Christine Loyd.

**Methodology:** Yue Zhang, Richard E. Kennedy, Norman R. Estes, Samantha Giordano-Mooga, Christine Loyd.

**Project administration:** Samantha Giordano-Mooga, Christine Loyd.

**Resources:** Richard E. Kennedy, Samantha Giordano-Mooga, Christine Loyd.

**Software:** Yue Zhang, Richard E. Kennedy.

**Supervision:** Norman R. Estes, Samantha Giordano-Mooga, Christine Loyd.

**Validation:** Richard E. Kennedy.

**Visualization:** Ashley Guy, Shannon McAuliffe, Robbie Cross, Christine Loyd.

**Writing – original draft:** Ashley Guy, Shannon McAuliffe, Christine Loyd.

**Writing – review & editing:** Ashley Guy, Shannon McAuliffe, Robbie Cross, Yue Zhang, Richard E. Kennedy, Norman R. Estes, Samantha Giordano-Mooga, Christine Loyd.

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
