## [Decision Letter · Decision Letter 0]

1 Apr 2024

PONE-D-24-05281Pilot study assessing gut microbial diversity among sexual and gender minority young adultsPLOS ONE

Dear Dr. Loyd,

Thank you for submitting your manuscript to PLOS ONE. After careful consideration, we feel that it has merit but does not fully meet PLOS ONE’s publication criteria as it currently stands. Therefore, we invite you to submit a revised version of the manuscript that addresses the points raised during the review process.

Reviewers have expressed concerns regarding the decision to combine sexual and gender minorities in this study, citing the potential variability that could impact the gut microbiome. It is important for the authors to address these concerns and provide a response.

We look forward to receiving your revised manuscript.

Kind regards,

Suresh Pallikkuth

Academic Editor

PLOS ONE

Reviewers' comments:

Reviewer's Responses to Questions

**Comments to the Author**

1. Is the manuscript technically sound, and do the data support the conclusions?

Reviewer #1: Yes

Reviewer #2: Partly

2. Has the statistical analysis been performed appropriately and rigorously? 

Reviewer #1: Yes

Reviewer #2: Yes

3. Have the authors made all data underlying the findings in their manuscript fully available?

Reviewer #1: No

Reviewer #2: No

4. Is the manuscript presented in an intelligible fashion and written in standard English?

Reviewer #1: Yes

Reviewer #2: Yes

5. Review Comments to the Author

Reviewer #1: Overall, study is important contribution to literature. There are some modifications required. Specific to questions above, they state in results that some of their data are not presented. could consider including as an appendix

Reviewer #2: In manuscript PONE-D-24-05281, Guy et al, characterize the gut microbial diversity of SGM and compare them to that of CIS-HET population, to understand if gut microbiome is a predictor of minority-related stress. While it is crucial to study gut microbiome as a contributor of disease outcomes in the SGM community, this manuscript needs clarification in the following areas.

1. How do the authors conclude if the differences in gut microbiome between SGM and CIS-HET were not due to any non-stress-related disease conditions, that were not self-reported, such as obesity, hypertension etc?

2. The authors should measure beta diversity to truly interpret differences in microbiome between SGM and CIS-HET.

3. Since the identification of CIS-HET was self-reported, the authors should include in the limitation a possibility of missing SGM individuals who were not ready to declare their identity yet.

4. Since the participants were from multiple races, the authors need to discuss if that would impact the study outcome?

5. Can the authors clarify why cortisol in the saliva and nails were used to assess stress, as opposed to other cortisol independent pathways?

6. Can the authors estimate the differential bacterial class, order, family, genus and species between SGM and CIS-HET groups?

6. PLOS authors have the option to publish the peer review history of their article (what does this mean?). If published, this will include your full peer review and any attached files.

Reviewer #1: No

Reviewer #2: **Yes: **Ria Goswami

---

## [Author Response · Author response to Decision Letter 0]

25 Apr 2024

April 25, 2024

 PLOS ONE 

REF: Manuscript #: PONE-D-24-05281

Dear Dr. Pallikkuth-

Thank you for your email of April 1, 2024 and for the opportunity to complete a revision of our manuscript entitled “Pilot study assessing gut microbial diversity among sexual and gender minority young adults”. Below, please find our item-by-item response to reviewer comments. The pages, paragraphs, and line numbers in the letter refer to locations in the revised manuscript which is provided in the clean and tracked changes formats. We added continuous line numbering on the tracked version to assist in reviewing the changes we made on the manuscript. The line call outs in the response letter relate to the lines in the tracked version of the revised manuscript. We hope these changes are satisfactory and we look forward to your response.

Academic Editor Comments:

Reviewers have expressed concerns regarding the decision to combine sexual and gender minorities in this study, citing the potential variability that could impact the gut microbiome. It is important for the authors to address these concerns and provide a response.

Response: Thank you for providing this feedback. We agree that this should be addressed in the text. We have added the following text to the discussion to address this comment. Discussion section, line 311-316, added text is in red: “Finally, a convenience sample was selected and the sample size was small for this pilot study, thus our analysis combined all SGM members into a single group which limited our ability to differentiate impacts on the gut microbiome among distinct sexual and gender minority groups. Therefore, a larger more comprehensive study focused on replicating findings presented herein, identifying differences between SGM sub-groups, and determining the role of non-stress related health conditions on the microbiome is needed.”

Journal Requirements:

1. When submitting your revision, we need you to address these additional requirements. Please ensure that your manuscript meets PLOS ONE’s style requirements, including those for file naming.

Response: We have revised the manuscript to ensure it meets all formatting/style requirements for PLOS ONE using the template links provided. 

Response: We have corrected this so the information is the same across both sections.

3. We note that you have indicated that there are restrictions to data sharing for this study. PLOS only allows data to be available upon request if there are legal or ethical restrictions on sharing data publicly. 

Response: We have deposited de-identified gut microbiome data in Figshare for the CIS-HET and SGM groups (DOI 10.6084/m9.figshare.25639596). We have added text to the results section, line 242-244: “Report of the taxonomic breakdown of microbial differences between the CIS-HET and SGM groups can be found in a public dataset (DOI 10.6084/m9.figshare.25639596).” Since the microbiome data is not self-reported data, the authors believe that this data is less likely to be linked back to individual participants of this small pilot study so it can be made publicly available. Alternatively, due to the limited sample size of this small pilot study, it is unethical to publicly share self-reported data due to the increased likelihood of linking the data back to the individuals enrolled in this study. Furthermore, because of the unique social experience of the SGM community that includes multiple levels of discrimination in the US South, it is vital that we keep the identities of our participants confidential. De-identified self-reported data is available only upon request from the office of the Research Director in the UAB Clinical and Diagnostic Sciences (CDS) Department. The current point of contact is Keith McGregor, email: kmmcgreg@uab.edu. The limited dataset will remain with the Research Director’s office moving forward and this email address can be contacted for external requests for data access for the foreseeable future. The corresponding author will respond to the journal regarding changes to the email address for data access requests in the long-term. The CDS Research Director’s office is developing centralized email address to ensure continued ability to making data access requests. 

Response: We have added the data to the text to address this issue. Results section, line 202-206: “In the CIS-HET group, 35 participants (81.4%) reported using neuropsychological medications regularly, while 18 (81.8%) in the SGM group reported using this type of medication regularly (p=0.376). Further, 8 participants (18.6%) in the CIS-HET group and 4 (18.2%) in the SGM group reported using hormonal medications regularly (p=1.000).”

Reviewer's Responses to Questions

Comments to the Author

1. Is the manuscript technically sound, and do the data support the conclusions?

Reviewer #1: Yes

Reviewer #2: Partly

Response: We have addressed reviewer comments below which we hope will address Reviewer #2’s concerns.

2. Has the statistical analysis been performed appropriately and rigorously?

Reviewer #1: Yes

Reviewer #2: Yes

3. Have the authors made all data underlying the findings in their manuscript fully available?

Reviewer #1: No

Reviewer #2: No

Response: Please see response to “Journal Requirements, #3” above. 

4. Is the manuscript presented in an intelligible fashion and written in standard English?

Reviewer #1: Yes

Reviewer #2: Yes

5. Review Comments to the Author

Reviewer #1: Overall, study is important contribution to literature. There are some modifications required. Specific to questions above, they state in results that some of their data are not presented. could consider including as an appendix

Response: We thank the reviewer for this comment. We have removed the “data not shown” verbiage and included the data in the body of the manuscript. Please see results section, line 202-206: “In the CIS-HET group, 35 participants (81.4%) reported using neuropsychological medications regularly, while 18 (81.8%) in the SGM group reported using this type of medication regularly (p=0.376). Further, 8 participants (18.6%) in the CIS-HET group and 4 (18.2%) in the SGM group reported using hormonal medications regularly (p=1.000).” 

Reviewer #2: In manuscript PONE-D-24-05281, Guy et al, characterize the gut microbial diversity of SGM and compare them to that of CIS-HET population, to understand if gut microbiome is a predictor of minority-related stress. While it is crucial to study gut microbiome as a contributor of disease outcomes in the SGM community, this manuscript needs clarification in the following areas.

1. How do the authors conclude if the differences in gut microbiome between SGM and CIS-HET were not due to any non-stress-related disease conditions, that were not self-reported, such as obesity, hypertension etc?

Response: We thank the reviewer for this question. Since this was a pilot study, we collected a limited dataset. Yet, we agree with the reviewer that this information would provide some indication of what is driving the difference in gut microbial diversity between the CIS-HET and SGM groups. Future more comprehensive studies will investigate the role of non-stress related conditions in this difference. To this point, we added text to the limitations section of the discussion, please see Lines 314-316: “a larger more comprehensive study focused on replicating findings presented herein, identifying differences between SGM sub-groups, and determining the role of non-stress related health conditions on the microbiome is needed.” 

2. The authors should measure beta diversity to truly interpret differences in microbiome between SGM and CIS-HET.

Response: We thank the reviewer for this comment. We have completed the beta diversity analysis and provided this data in the text of the manuscript. Text has been added to the methods section, lines 174-178: “Variability of microbial community composition among samples (beta diversity) from participants in each group was analyzed using the Bray Curtis method, the unweighted uniFrac and the weighted uniFrac methods. The uniFrac measurement of beta diversity uses phylogenetic information to compare samples to identify differences among microbial communities for each group.[34] See the public dataset at DOI 10.6084/m9.figshare.25639596.” Text has also been added to the Results section, lines 233-236:” Beta diversity analysis between the CIS-HET and SGM groups identified a significant difference between groups using the unweighted uniFrac method (p=0.041). However, no difference was observed using the Bray Curtis and the weighted uniFrac (p=0.081 and 0.645 respectively).” Text was also added to the discussion section, lines 277-279: “Another key finding was that the unweighted uniFrac measure of beta diversity was significantly different between the CIS-HET and the SGM groups. These data suggest some differences in the microbial communities present among the CIS-HET and SGM groups.”

3. Since the identification of CIS-HET was self-reported, the authors should include in the limitation a possibility of missing SGM individuals who were not ready to declare their identity yet.

Response: We agree with the reviewer, it is important to note this limitation. We added text to the limitation section of the discussion. Please see Lines 307-308: “Furthermore, it is possible that some SGM individuals were missed because they were not ready to self-report their sexual and/or gender minority identity.”

4. Since the participants were from multiple races, the authors need to discuss if that would impact the study outcome?

Response: We thank the reviewer for this question. Since race is known to impact the gut microbiome, we assessed group differences in terms of race and identified no significant difference. This finding is reported as part of Table 1. See lines 195-196 in the results section, this text was part of the original manuscript.

5. Can the authors clarify why cortisol in the saliva and nails were used to assess stress, as opposed to other cortisol independent pathways?

Response: We thank the reviewer for this question. The cortisol pathway was selected due to ease of measuring acute and chronic levels in saliva and nails respectively. However, we agree that other pathways of stress need to be assessed to determine their role in impacting gut microbiome health in the SGM and CIS-HET groups. We have added text to the discussion section, line 302-303: “Future research should determine the role of cortisol-independent pathways of stress on gut microbiome health among the SGM community.”

6. Can the authors estimate the differential bacterial class, order, family, genus and species between SGM and CIS-HET groups?

Response: We have provided this data publicly, which provides the differential bacterial class, order, family, genus, species between groups. 

Figshare link: (DOI 10.6084/m9.figshare.25639596)

---

## [Decision Letter · Decision Letter 1]

23 May 2024

PONE-D-24-05281R1Pilot study assessing gut microbial diversity among sexual and gender minority young adultsPLOS ONE

Dear Dr. Loyd,

Thank you for submitting your manuscript to PLOS ONE. After careful consideration, we feel that it has merit but does not fully meet PLOS ONE’s publication criteria as it currently stands. Therefore, we invite you to submit a revised version of the manuscript that addresses the points raised during the review process.

Presenting sexual and gender minorities as one group is still a concern. The authors need to satisfactorily address this comment.

We look forward to receiving your revised manuscript.

Kind regards,

Suresh Pallikkuth

Academic Editor

PLOS ONE

Reviewers' comments:

Reviewer's Responses to Questions

**Comments to the Author**

1. If the authors have adequately addressed your comments raised in a previous round of review and you feel that this manuscript is now acceptable for publication, you may indicate that here to bypass the “Comments to the Author” section, enter your conflict of interest statement in the “Confidential to Editor” section, and submit your "Accept" recommendation.

Reviewer #1: (No Response)

Reviewer #2: All comments have been addressed

2. Is the manuscript technically sound, and do the data support the conclusions?

Reviewer #1: Yes

Reviewer #2: Yes

3. Has the statistical analysis been performed appropriately and rigorously? 

Reviewer #1: Yes

Reviewer #2: Yes

4. Have the authors made all data underlying the findings in their manuscript fully available?

Reviewer #1: Yes

Reviewer #2: Yes

5. Is the manuscript presented in an intelligible fashion and written in standard English?

Reviewer #1: Yes

Reviewer #2: Yes

6. Review Comments to the Author

Reviewer #1: I appreciate the modifications made by the authors but still have reservations presenting sexual and gender minorities as one group. They cannot be grouped together as these are extremely different minority groups. I undertand that this is a pilot and thus a small dataset, but it is not acceptable to consider these as one group.

Reviewer #2: I thank the authors for clarifying and addressing all my questions in a very clear and well understandable fashion.

7. PLOS authors have the option to publish the peer review history of their article (what does this mean?). If published, this will include your full peer review and any attached files.

Reviewer #1: No

Reviewer #2: **Yes: **Ria Goswami

---

## [Author Response · Author response to Decision Letter 1]

28 May 2024

May 28, 2024

 PLOS ONE 

REF: Manuscript #: PONE-D-24-05281

Dear Dr. Pallikkuth-

Thank you for your email of May 23, 2024. We have edited our manuscript and addressed the comment from reviewers below. We hope this revision is satisfactory and we look forward to hearing from you again soon. We have unpublished our raw data in figshare and will republish it making it public once again once it is accepted for publication. The DOI will remain the same: 10.6084/m9.figshare.25639596.

Reviewer comment:

(1) Presenting sexual and gender minorities as one group is still a concern. The authors need to satisfactorily address this comment

a. Response: We agree that examining differences between gender minorities and sexual minorities is important. We have addressed completed additional analyses and determined that there was no difference between sexual minorities only and gender/sexual minorities (all gender minorities in this study also self-identified as sexual minorities) in terms of demographics and outcomes. 

We have addressed this in the text.

Line 195-197 in the results: When the SGM group was separated into sexual minority only (N=14) and gender/sexual minorities (N=8), there was no significant difference between groups in demographics (age, race), psychological factors (PHQ-9, GAD-7, PSS-10), or nail and saliva cortisol levels measured. Also, lines 217-220 in the results: When sexual minorities only were analyzed separately from gender and sexual minorities compared to the CIS-HET group, alpha diversity was significantly reduced among sexual minorities only (N=14; Shannon Alpha Diversity mean=4.89, SD=0.76) and gender and sexual minorities (N=8; Shannon Alpha Diversity mean=4.69, SD=0.89) compared to the CIS-HET group.

We do agree that additional more robust and comprehensive studies need to be completed to understand the impact of gender minority only status on health of the gut microbiome. Thus, we have added text to the discussion to this point. See lines 297-298: Further, the SGM group did not include individuals identifying as lesbian nor did it include gender minority only individuals thus, findings do not apply to these minority groups.

---

## [Decision Letter · Decision Letter 2]

21 Jun 2024

Pilot study assessing gut microbial diversity among sexual and gender minority young adults

PONE-D-24-05281R2

Dear Dr. Loyd,

We’re pleased to inform you that your manuscript has been judged scientifically suitable for publication and will be formally accepted for publication once it meets all outstanding technical requirements.

Kind regards,

Suresh Pallikkuth

Academic Editor

PLOS ONE

Additional Editor Comments (optional):

Reviewers' comments:

Reviewer's Responses to Questions

**Comments to the Author**

1. If the authors have adequately addressed your comments raised in a previous round of review and you feel that this manuscript is now acceptable for publication, you may indicate that here to bypass the “Comments to the Author” section, enter your conflict of interest statement in the “Confidential to Editor” section, and submit your "Accept" recommendation.

Reviewer #1: All comments have been addressed

2. Is the manuscript technically sound, and do the data support the conclusions?

Reviewer #1: Yes

3. Has the statistical analysis been performed appropriately and rigorously? 

Reviewer #1: Yes

4. Have the authors made all data underlying the findings in their manuscript fully available?

Reviewer #1: Yes

5. Is the manuscript presented in an intelligible fashion and written in standard English?

Reviewer #1: Yes

6. Review Comments to the Author

Reviewer #1: (No Response)

7. PLOS authors have the option to publish the peer review history of their article (what does this mean?). If published, this will include your full peer review and any attached files.

Reviewer #1: No

---

## [Editor Report · Acceptance letter]

24 Jun 2024

PONE-D-24-05281R2 

PLOS ONE

Dear Dr. Loyd, 

I'm pleased to inform you that your manuscript has been deemed suitable for publication in PLOS ONE. Congratulations! Your manuscript is now being handed over to our production team.

Kind regards, 

on behalf of

Dr. Suresh Pallikkuth 

Academic Editor

PLOS ONE